# Genome-Wide Identification and Characterization of the Abiotic-Stress-Responsive *GRF* Gene Family in Diploid Woodland Strawberry (*Fragaria vesca*)

**DOI:** 10.3390/plants10091916

**Published:** 2021-09-15

**Authors:** Zhiqi Li, Qian Xie, Jiahui Yan, Jianqing Chen, Qingxi Chen

**Affiliations:** 1College of Horticulture, Fujian Agriculture and Forestry University, Fuzhou 350002, China; ZhiqiLi@fafu.edu.cn (Z.L.); xieqi0416@163.com (Q.X.); JiahuiYan@fafu.edu.cn (J.Y.); 2Horticultural Plant Biology and Metabolomices Center, Haixia Institute of Science and Technology, Fujian Agriculture and Forestry University, Fuzhou 350002, China

**Keywords:** GRF, gene family, strawberry, abiotic stress, expression pattern, cis-acting element

## Abstract

Growth regulatory factors (GRF) are plant-specific transcription factors that play an important role in plant resistance to stress. This gene family in strawberry has not been investigated previously. In this study, 10 *GRF* genes were identified in the genome of the diploid woodland strawberry (*Fragaria* *vesca*). Chromosome analysis showed that the 10 *FvGRF* genes were unevenly distributed on five chromosomes. Phylogenetic analysis resolved the FvGRF proteins into five groups. Genes of similar structure were placed in the same group, which was indicative of functional redundance. Whole-genome duplication/segmental duplication and dispersed duplication events effectively promoted expansion of the strawberry *GRF* gene family. Quantitative reverse transcription-PCR analysis suggested that *FvGRF* genes played potential roles in the growth and development of vegetative organs. Expression profile analysis revealed that *FvGRF3*, *FvGRF5*, and *FvGRF7* were up-regulated under low-temperature stress, *FvGRF4* and *FvGRF9* were up-regulated under high-temperature stress, *FvGRF6* and *FvGRF8* were up-regulated under drought stress, *FvGRF3*, *FvGRF6*, and *FvGRF8* were up-regulated under salt stress, *FvGRF2*, *FvGRF7*, and *FvGRF9* were up-regulated under salicylic acid treatment, and *FvGRF3*, *FvGRF7*, *FvGRF9*, and *FvGRF10* were up-regulated under abscisic acid treatment. Promoter analysis indicated that *FvGRF* genes were involved in plant growth and development and stress response. These results provide a theoretical and empirical foundation for the elucidation of the mechanisms of abiotic stress responses in strawberry.

## 1. Introduction

Strawberry (*Fragaria × ananassa*) is cultivated worldwide on account of its ornamental value and nutritional fruit. Strawberry is widely grown under a protected environment and China is among the largest strawberry producers in the world. Strawberry is a suitable model plant [1,2] for fruit research because of its small fruit size, short growth cycle, and high efficiency for genetic transformation. Therefore, strawberry can potentially make an important contribution to commercial production and scientific research in the fruit and vegetable industry. However, the cultivated hybrid strawberry is octaploid [3]; thus, its genetic background is highly complex. The diploid ancestral species, woodland strawberry (*Fragaria vesca*), is more amenable to genetic analysis. In addition, published strawberry genome sequences enable identification of the genetic basis of desirable agronomic traits and stress-resistance genes in strawberry at the genome level.

Given the environmental variability experienced during growth, plants are vulnerable to diverse abiotic stresses, such as low or high temperature, drought, and high salinity, which may adversely affect growth and development [4,5]. In response to exposure to adverse external variables, transcription factors (TFs) play an important role in the regulation of functional gene expression in response to growth and development and signal transduction under stress [6,7]. Growth-regulating factors (GRFs) are plant-specific proteins [8] that regulate growth and development and stress responses. GRF proteins contain two conserved domains (QLQ and WRC domains) in the N-terminal region [9,10]. The QLQ domain is similar to the N-terminus of SWI2/SNF2 in yeast and can bind with SNF11 to form a chromatin remodeling complex [11]. In addition, the QLQ domain can interact with the conserved structure of the GIF protein SNH to perform the function of transcriptional activation. The WRC domain contains a nuclear localization signal motif and a zinc finger motif that plays a role in DNA binding, which can regulate the expression of downstream genes [12] by binding to the cis-acting region of the target gene.

The GRF transcription factor family has been identified and analyzed at the genome level in many plant species; for example, nine GRF family members have been identified in *Arabidopsis* [13], 12 in rice [14], 35 in rapeseed [15], 17 in Chinese cabbage [16], 25 in tobacco [17], and 30 in wheat [18]. *GRF* genes play an important role in the plant response to abiotic stress [19,20]. The overexpression of *AtGRF7* results in up-regulation of the dehydrogenation response element binding protein 2A (DREB2A), which confers increased tolerance to salt and drought stress [21]. Under abiotic stress, *AtGRF7* directly binds to the *cis*-acting regulatory element TGTCAGG in the promoter of DREB2A to inhibit the expression of wild-type DREB2A, thus maintaining the growth rate [22,23]. Gene expression profile analysis of the *atgrf7-1* T-DNA insertion mutant showed that a large proportion of the up-regulated genes are associated with responses to stress and abscisic acid (ABA) [22]. The downstream targets of *AtGRF1* and *AtGRF3* are involved in defense response and disease resistance, which indicates that *AtGRF1* and *AtGRF3* play primary roles in the coordination of plant growth and defense signals [20]. Expression patterns of *GhGRF1A*, *GhGRF1D*, and *GhGRF17D* in cotton change in response to salt stress [24]. Under shade stress, all *GmGRF* genes are significantly down-regulated in soybean [25]. Tomato *SlGRF1*, *SlGRF4*, *SlGRF5*, *SlGRF7*, *SlGRF10*, *SlGRF11*, and *SlGRF12* each play an important role in ABA signal transduction [26].

In addition, GRF transcription factors are involved in plant growth and development, including root development [27], flowering [28,29], and leaf size and longevity [30]. The overexpression in Arabidopsis of *AtGRF1* and *AtGRF2* leads to cotyledon and leaf enlargement, and *AtGRF8* is involved in Arabidopsis flower development. The overexpression of *BnGRF2* increases seed weight and oil content in rapeseed by regulating the cell number and photosynthesis. Although the *GRF* family has been well characterized in model plants and other plant species, information on the function and evolutionary characteristics of the *GRF* gene family in strawberry remains limited.

Therefore, we studied the role of the *GRF* gene family in abiotic stress responses in strawberry. A reference genome for woodland strawberry (*F. vesca*, 2 *n* = 2 *x* = 14) has been published [31]. The genome comprises approximately 240 MB and presents an opportunity for the genome-wide mining of GRF transcription factors. Although the *GRF* gene family has been previously characterized in many plant species, little is known about the family in *F. vesca*. In this study, we conducted a genome-wide search and identified 10 members of the *GRF* gene family in the woodland strawberry genome. The functions of *FvGRFs* in response to abiotic stresses (low temperature, high temperature, drought, and salinity) and hormone treatment (salicylic acid and abscisic acid) were explored. In addition, bioinformatics analyses were conducted to provide insights into the structure and function, as well as the evolution, of the *FvGRF* genes. The results provide a foundation for further investigation of the functions of strawberry *GRF* genes in response to abiotic stresses.

## 2. Results

### 2.1. Phylogenetic Analysis of the GRF Family

To identify the *GRF* gene sequence of strawberry, *GRF* candidate genes were searched from the *Fragaria vesca* genome according to two strategies: Hidden Markov Model search (HMM search) using the HMM profiles PF08880 (QLQ domain) and PF08879 (WRC domain); BLASTP search using GRF proteins from *Arabidopsis* as queries. As a result, ten members of the *FvGRF* gene family were identified from the diploid woodland strawberry genome (V4a) (Appendix A). The genes were designated *FvGRF1* to *FvGRF10* according to their homologous relationship with *Arabidopsis thaliana* genes. To explore the functions of the *FvGRF* family members, a phylogenetic analysis of GRF protein sequences from *Arabidopsis*, rice, woodland strawberry, poplar, maize, Chinese cabbage, and rapeseed was performed. The strawberry GRF proteins were resolved into five clades, herein designated subfamilies A to E, of which *FvGRF1*, *FvGRF5*, *FvGRF7*, and *FvGRF10* belonged to subfamily A, *FvGRF6* belonged to subfamily B, *FvGRF4* and *FvGRF9* belonged to subfamily C, *FvGRF2* and *FvGRF3* belonged to subfamily D, and *FvGRF8* belonged to subfamily E (Figure 1a).

Multiple sequence alignment and conserved domain analysis of the amino acid sequences of the *FvGRF* family members revealed that all *FvGRF* proteins contained QLQ and WRC domains. *FvGRF4* and *FvGRF8* also contained a second WRC domain downstream of the first WRC domain. In addition, the zinc finger motif (CCCH) [32] was observed in the WRC domain of all identified *FvGRF* proteins (Appendix A).

Analysis of the gene structure and conserved motifs supported the phylogenetic reconstruction of the *GRF* gene family. *FvGRF1*, *FvGRF5*, and *FvGRF7* in subfamily A contained three introns, whereas *FvGRF10* contained nine introns, the members of subfamilies B, C, and D contained four introns, and the members of subfamily E contained two introns (Appendix A). To identify potential conserved motifs, the MEME tool was used to analyze the sequences of the 10 *FvGRF* genes. Ten conserved motifs were detected, which were designated motif1 to motif10. Among these motifs, each gene contained not only the QLQ and WRC motifs, but also 2–5 additional conserved motifs. Interestingly, motif2, motif4, and motif7 were observed only in subfamily A members, motif8 only in subfamily D and E members, and motif9 only in subfamily D members. These specific motifs may contribute to the functional diversity of *GRF* genes from different subfamilies (Figure 1b).

### 2.2. Chromosomal Location, Gene Structure, and Conserved Motif Analysis

The *FvGRF* genes were not evenly distributed on all chromosomes. The 10 *FvGRF* genes were distributed on five chromosomes. *FvGRF1* and *FvGRF2* were located on Chrom01, *FvGRF3* was located on Chrom02, *FvGRF4* and *FvGRF5* were located on Chrom05, *FvGRF6* and *FvGRF7* were located on Chrom06, and *FvGRF8*, *FvGRF9*, and *FvGRF10* were located on Chrom07. Thus, Chrom03 and Chrom04 did not carry *FvGRF* genes (Figure 2).

### 2.3. Different Duplication Events Control the Expansion of GRF Genes in Strawberries and Arabidopsis Thaliana

To gain further insight into the evolution of strawberry *GRF* genes, we analyzed the duplication events of *GRF* genes in *Arabidopsis thaliana* and strawberry. Dispersed duplication was the primary mode of *GRF* gene replication in *Arabidopsis*, accounting for 55.56% (five of nine), singleton genes accounted for 11.11% (one of nine), and 33.33% of the genes involved WGD/segmental duplication (three of nine) (Figure 3). In contrast, WGD/segmental duplication was the primary mode of replication of strawberry *GRF* genes, accounting for 60% (six out of ten), and genes involved in dispersed duplication events and singleton genes accounted for 20% each (two out of ten) (Appendix A). Proximal duplication and tandem duplication events were not detected in the *Arabidopsis* and strawberry *GRF* families. These results suggested that different gene duplication events controlled expansion of the *GRF* family in *Arabidopsis* and strawberry (Appendix A).

### 2.4. Expression Patterns of FvGRF Genes in Different Organs

To study the expression patterns of *GRF* genes in different organs of diploid woodland strawberry, qRT-PCR was used to quantify the expression level of *FvGRF* genes in the root, stem, leaf, flower, and fruit. Certain *FvGRF* genes were highly expressed in specific organs, whereas other *FvGRF* genes showed similar expression patterns in different organs (Figure 3), which may be indicative of functional differences of the genes in strawberry growth and development. For example, *FvGRF3*, *FvGRF4*, and *FvGRF7* were expressed at relatively high levels in the root, whereas *FvGRF2* and *FvGRF9* were mainly expressed in vegetative organs (Figure 4). *FvGRF5* and *FvGRF10* were highly expressed in different tissues. *FvGRF1* was more highly expressed in the stem and flower. The expression level of *FvGRF8* was higher in the root, stem, and fruit. Except for *FvGRF1*, the expression level of *FvGRF* genes in vegetative organs was generally higher than that in reproductive organs, which indicated that *FvGRF* genes may play important roles in the growth and development of vegetative organs.

### 2.5. Expression Pattern of FvGRF Genes under Stress Treatments

To explore the potential role of *FvGRF* genes in plant responses to various environmental stresses, the expression level of the 10 *FvGRF* genes under low temperature, high temperature, drought, and salt stress was determined by qRT-PCR analysis. Generally, the *FvGRF* genes differed in the degree of response to low temperature, high temperature, drought, and salt stress. Compared with the high temperature and drought treatments, the *FvGRF* genes were more highly responsive to low temperature and salt stress (Figure 4). The expression of *FvGRF3*, *FvGRF5*, and *FvGRF7* was up-regulated under low-temperature stress, *FvGRF4* and *FvGRF9* were up-regulated under high-temperature stress, *FvGRF6* and *FvGRF8* were up-regulated under drought stress, *FvGRF3*, *FvGRF6*, and *FvGRF8* were up-regulated under salt stress, and the other genes were down-regulated to varying degrees. The degree of response of *FvGRF* genes to low-temperature stress was similar to that under salt stress.

The plant hormones SA and ABA play important roles in plant stress signal response and in plant growth and development. To evaluate if *FvGRF* gene expression was induced in response to plant hormone treatment, the expression patterns of the *FvGRF* genes in the leaf in response to exogenous SA and ABA treatment were analyzed by qRT-PCR. In response to SA treatment, the expression levels of *FvGRF2*, *FvGRF7*, and *FvGRF9* were significantly increased to a high level from 4 h after treatment, and were maintained at a high expression level during the entire experimental period, whereas the other genes were up-regulated or down-regulated in certain periods (Figure 5). Under ABA treatment, *FvGRF3*, *FvGRF7*, *FvGRF9*, and *FvGRF10* were up-regulated to varying degrees, whereas all other *FvGRF* genes were down-regulated or almost unchanged. It is notable that *FvGRF7* and *FvGRF9* were up-regulated in response to treatment with SA and ABA (Figure 4).

### 2.6. Identification of Cis-Acting Regulatory Elements in the Promoter of FvGRF Genes

To explore the regulation of strawberry GRF family members, analysis of the *cis*-acting regulatory elements in the 1.5 kb region upstream of the initiation codon of all *FvGRF* family members was conducted using the PlantCARE online portal. The GCN4 motif [33], a *cis*-acting regulatory element associated with endosperm expression, was detected in *FvGRF3*, *FvGRF5*, *FvGRF9*, and *FvGRF10* (Figure 6). It was noteworthy that the element involved in the regulation of circadian rhythms [34] was detected in *FvGRF9*, and only *FvGRF1* contained an element involved in mechanical injury response (WUN motif). In addition, the gliadin metabolic regulatory element (O2-site), meristem expression and specific activation elements (CAT-box and CCGTCC-box), and seed-specific regulatory element (RY-element) [35] were identified in the promoter of *FvGRF* genes. With regard to hormone-related *cis*-acting elements, the SA response element (TCA-element) [36], methyl jasmonate response elements (CGTCA-motif and TGACG-motif) [37], and ABA response element (ABRE) [38] were identified in the promoters of two, seven, and 10 *FvGRF* genes, respectively. Gibberellin response elements (GARE-motif and P-box) [39] and auxin response elements (TGA-element and AuxRR-core) [40] were observed in five and three *FvGRF* genes, respectively. A large number of elements associated with hormone response were identified in the *FvGRF* promoter sequence, which indicated that plant hormones may play a crucial role in regulating the functions of *FvGRF* genes in plant growth and development (Figure 6B). In addition, several *cis*-acting elements associated with response to stresses (such as drought, extreme temperature, and salinity) were observed in the promoter region of *FvGRF* genes (Appendix A).

## 3. Discussion

Growth regulatory factors are plant-specific transcription factors. Previous studies have shown that GRFs play an important role in coordinating growth under stress. The present study aimed to identify candidate genes involved in stress regulation among members of the strawberry *GRF* family. Using the diploid woodland strawberry reference genome (V4a version), 10 members of the *FvGRF* gene family were identified in this study. Whole-genome duplication/segmental duplication was the primary driving force for expansion of the strawberry *GRF* family. Strawberry *GRF* genes were predominantly expressed in vegetative organs and expression levels changed to varying degrees in response to low temperature, high temperature, drought, and salinity stress, and to exogenous SA and ABA treatment. These results indicated that *FvGRF* genes showed potential regulatory functions in abiotic stress responses.

Gene duplication plays an important role in gene family evolution and is the primary mechanism for the generation of novel evolutionary innovations, such as tandem duplication and WGD/segmental duplication [41]. The present results indicated that dispersed duplication and WGD/segmental duplication effectively promoted expansion of the *GRF* gene family in *Arabidopsis* and strawberry, but tandem duplication events were not detected in either gene family. This phenomenon has previously been reported [8]. A gene family may show a common nonrandom origin pattern and a conserved duplication pattern in different species [42]. However, in the present study, decentralized duplication was the predominant driving force in *Arabidopsis*, whereas WGD/segmental duplication was the primary driving force in strawberry. These results indicated that the main duplication patterns of the *Arabidopsis* and strawberry *GRF* gene families were not always strictly conserved and that nonrandom patterns from different sources were common.

The expression patterns of the *FvGRF* genes were analyzed to evaluate their potential roles in different organs of strawberry. Six *FvGRF* genes were most highly expressed in the roots. Previous studies have shown that GRF proteins may play an important role in plant root development or physiological processes [27]. In *Arabidopsis*, *AtGRF1* and *AtGRF3* are highly expressed in the roots [20]. In the present study, *FvGRF2* and *FvGRF3* were shown to be homologous with *AtGRF1* and were highly expressed in the roots, suggesting that members of this subfamily perform a potentially conserved function in the roots. In rice, *OsGRF10* is highly expressed in the leaves [14]. In the present study, we observed that *FvGRF5* and *FvGRF10* were homologous with *OsGRF10* and were also highly expressed in the leaves. Thus, members of this subfamily may show a potentially conserved function in the leaves. *GRF* genes are usually more highly expressed in actively growing tissues than in mature tissues [43]. Overall, the expression level of *FvGRF* genes in vegetative organs was higher than that in reproductive organs, which suggested that *FvGRF* genes may play important roles in the growth and development of vegetative organs.

The ABA signaling pathway is considered to be a central regulator of abiotic stress in plants and mediates the expression of stress-resistance genes. The ABRE is the main *cis*-acting element involved in ABA-responsive gene expression. Among strawberry *GFR* genes, *FvGRF3* attained the highest expression level at 12 h after ABA treatment, *FvGRF7* showed the highest level at 48 h, *FvGRF9* attained the highest level at 12 h, and *FvGRF10* was the most highly expressed gene at 48 h and was up-regulated by 10.56-fold. All *FvGRF* genes contained an ABRE in the promoter. The presence of multiple ABRE copies can provide ABA with the ability to respond to the promoters, whereas a single-copy ABRE is not responsive to ABA [44]. In the present study, we observed that *FvGRF7* and *FvGRF9* contained two ABRE-acting elements, which may be the reason for their up-regulation in response to exogenous ABA. However, only one ABRE is present in the promoter of *rd29A* and there is no known coupling element, but exogenous ABA strongly induces *rd29A* expression, which indicates that a potential *cis*-acting regulatory element may act as a coupling element in the ABA response [45]. We identified only one ABRE in the promoters of *FvGRF3* and *FvGRF10*. It is speculated that other motifs may replace the function of the ABRE or coupling element, thereby resulting in the up-regulation of *FvGRF3* and *FvGRF10* in response to exogenous ABA.

Similarly, SA plays an important role in the regulation of plant responses to abiotic stress. Under SA treatment, *FvGRF2*, *FvGRF7*, and *FvGRF9* were up-regulated. Among these genes, *FvGRF2* attained the highest expression level at 4 h after SA treatment, which represented up-regulation by almost 16-fold, whereas *FvGRF7* showed the highest expression level at 4 h after SA treatment, which represented up-regulation by almost 17-fold. Promoter sequence analysis predicted the presence of the SA-responsive TCA-element in *FvGRF2* and *FvGRF7* but not in *FvGRF9*. Previous studies have shown that SA can induce *RLK* gene expression because a TTGCA sequence is present upstream of the *RLK* gene, which plays an important role in inducing the expression of many plant defense-related genes [46]. We observed the TTGCA sequence in the upstream region of *FvGRF9*, which may be an important reason for the up-regulation of *FvGRF9* expression in response to SA treatment. However, additional experimental evidence is needed to elucidate the transcriptional regulation of *FvGRF* genes in strawberry.

Drought and high salt stress strongly impact on plant growth and productivity. These stresses induce the expression of many genes in different plants. Various *cis*-acting elements in the stress response promoter play an important role in plant adaptation to environmental stress. A dehydration response element (DRE; TACCGACAT) responsible for dehydration and high salt-induced gene expression is present in the promoter region of *rd29A* (Narusaka Y et al., 2003). In the present study, expressions of *FvGRF6* and *FvGRF8* were up-regulated under drought and salt stress, and a DRE was observed in the promoter sequence of each gene. It is speculated that the DRE may be the reason for the up-regulated expression of *FvGRF6* and *FvGRF8* in response to salt and drought stress. Previous studies have shown that GT-1 components directly control the up-regulation of *OsRAV2* in response to high salinity [47]. Two salt stress response elements for *FvGRF3* were identified upstream of the GT-1 (GAAAAA) promoter. It is speculated that this element may account for the up-regulation of *FvGRF3* expression under salt stress.

In addition, plants are vulnerable to temperature fluctuations. High- and low-temperature stresses lead to plant stunting and reduce crop yields. Therefore, it is important to improve the ability of plants to resist low- and high-temperature stress. In *Arabidopsis*, *AtGRF7* can be used as a transcriptional activator of DREB2A and other stress-responsive genes to increase tolerance to high-temperature stress [22]. In the current study, *FvGRF4* and *FvGRF9* were up-regulated under high-temperature stress. Phylogenetic analysis resolved *AtGRF7*, *FvGRF4*, and *FvGRF9* in subfamily C, which suggested that this subfamily may perform a potentially conserved function under high-temperature stress. In the promoter of the *AtHsp90-1* gene, interaction of the CCAAT-box and stress response elements (STREs) increase *AtHsp90-1* expression in response to heat shock [48]. In the present study, the upstream region of the *FvGRF4* promoter was observed to contain CCAAT-box and STRE elements. We speculate that these elements may account for the up-regulation of *FvGRF4* expression in response to high-temperature stress. Analysis of *FvGRF* expression under low-temperature stress revealed that *FvGRF3*, *FvGRF5*, and *FvGRF7* were up-regulated in response to low-temperature treatment. The barley *blt4.9* gene promoter contains the CCGAAA sequence. Mutation analysis showed that deletion of the CCGAAA motif reduces the basic level of response to low temperature, which indicates that the CCGAAA motif plays an important role in the low-temperature stress response in barley [49]. In the present study, CCGAAA-motif elements were observed in the *FvGRF7* promoter and may explain the up-regulation of *FvGRF7* under low-temperature stress. In addition, *FvGRF3* and *FvGRF5* were up-regulated in response to low-temperature treatment, but no known cold-responsive regulatory elements were identified in the promoter of either gene. We speculate that the upregulation of *FvGRF3* and *FvGRF5* expression may be due to the presence of novel motifs in the promoter that may be the crucial elements in their response to low-temperature stress.

Different environmental factors affect gene expression, and gene expression also requires the coordination of inducible *cis*-acting regulatory elements and transcription factors in the promoter of environment-responsive genes. The type, number, and location of these elements may affect the level of gene expression. Therefore, further research is needed to gain an improved understanding of transcriptional regulatory mechanisms in strawberry, including transcription factors and their specific *cis*-acting regulatory elements.

## 4. Materials and Methods

### 4.1. Plant Materials, Growth Conditions, and Stress Treatments

Seeds of wild diploid woodland strawberry were obtained from the strawberry germplasm resource nursery of the College of Horticulture, Fujian Agriculture and Forestry University (26°10′ N, 119°23′ E), Fuzhou, China. The seeds were sown on Murashige and Skoog medium and then transferred to soil for subsequent growth after germination. The growth environment was 22 °C, relative humidity 75%, and a photoperiod of 13 h/11 h (day/night). As materials for gene expression analysis in different organs of woodland strawberry plants, samples of the roots, stems, leaves, flowers (at anthesis), and fruits (at the red fruit stage) were collected during reproductive growth. Each sample contains 3 replicates, with each replicate including 3 plants. All of the collected samples were snap-frozen in liquid nitrogen and kept at −80 °C until further use.

Four-month-old seedlings of uniform growth were selected for abiotic stress treatment. A solution of 400 mM/L of NaCl was sprayed onto potted plants to induce salt stress for 12 h. Potted plants were transferred to a growth room maintained at either 4 °C or 42 °C for 12 h for treatment with low-temperature and high-temperature stress, respectively. Plants were prepared by withholding water for 12 h of drought treatment. In addition, leaves were evenly sprayed with 40 μmol/L of ABA or 40 μmol/L of SA solution. The leaves of treated plants were sampled at 0, 3, 6, and 12 h after initiation of low temperature, high temperature, drought, and salt stress. The leaves of plants treated with SA or ABA were sampled at 0, 4, 8, 12, 16, 20, 24, and 48 h after treatment. Six leaves from three individual seedlings were selected and combined as one sample. All treatments were evaluated in triplicate. All collected samples were immediately wrapped in tin foil, frozen in liquid nitrogen, and stored at −80 °C until use.

### 4.2. Genome-Wide Identification of GRF Genes

The genome sequence, coding sequence, and amino acid sequence of the diploid strawberry (*Fragaria vesca*) genome assembly v4.0.a1 were downloaded from the Genome Database for Rosaceae (https://www.rosaceae.org (accessed on 16 March 2021)). The hidden Markov models of the characteristic GRF protein domains QLQ (PF08880) and WRC (PF08879) were downloaded from the Pfam database [50] (http://pfam.xfam.org/ (accessed on 16 March 2021)). Members of the *GRF* family in the complete strawberry genome were identified using HMMER software under the condition of *E*-value < 1 × 10^−10^. The amino acid sequences of *Arabidopsis GRF* family members were compared with those in the strawberry genome database using the BLASTP tool. The CD-Search tool (https://www.ncbi.nlm.nih.gov (accessed on 16 March 2021)) was used to verify the protein domains (Appendix A). The ExPASy ProtParam tool (https://web.expasy.org/protparam/ (accessed on 17 March 2021)) was used to predict the deduced protein sequence length, molecular weight, isoelectric point, and instability coefficient of strawberry *GRF* family members.

### 4.3. Phylogenetic Analysis

The GRF protein sequences of *Arabidopsis thaliana* were downloaded from The *Arabidopsis* Information Resource (https://www.arabidopsis.org (accessed on 16 March 2021)), those of rice from the Rice Genome Annotation Project (http://rice.plantbiology.msu.edu/analyses_search_locus.shtml (accessed on 16 March 2021)), those of Chinese cabbage from the Brassica database (http://brassicadb.org/brad/ (accessed on 16 March 2021)), those of rapeseed from BrassicaDB (http://brassicadb.org/brad/searchGene.php (accessed on 16 March 2021)), those of poplar and maize from Phytozome (https://phytozome.jgi.doe.gov/pz/portal.html (accessed on 16 March 2021)), and those of strawberry from the Genome Database for Rosaceae (https://www.rosaceae.org (accessed on 16 March 2021)). A multiple sequence alignment of GRF protein sequences of *Arabidopsis*, rice, rapeseed, poplar, maize, and strawberry was generated using ClustalX software [51] (Appendix A). A phylogenetic tree was constructed using the maximum likelihood method with MEGA6.0 software [40]. Support for the topology was assessed by performing a bootstrap analysis with 1000 replicates. The phylogenetic tree was visualized and edited using the iTOL online tool [52] (https://itol.embl.de/itol.cgi (accessed on 18 March 2021)).

### 4.4. Chromosomal Location, Gene Structure, and Conserved Sequence Analysis

The gff3 annotation file for the woodland strawberry genome was downloaded from the Genome Database for Rosaceae and the chromosomal locations of the *FvGRF* genes were extracted. TBtools [53] was used to visualize the chromosomal location of the *FvGRF* genes. The GSDS 2.0 online server [54] (https://gsds,cbi.pku.edu.cn/ (accessed on 20 March 2021)) was used to visualize the exon and intron structure of each gene. The amino acid sequence of each GRF protein of strawberry was submitted to the MEME online analysis website [55] (http://meme-suite.org/tools/meme (accessed on 20 March 2021)) to identify conserved protein motifs. The optimized MEME parameters were as follows: minimum pattern width 6; maximum pattern width 100; and use of the maximum number of programs, including up to 10. The conserved sequences were visualized using Adobe Illustrator 2020 software.

### 4.5. Synteny Analysis

Based on the method used in the Plant Genome Duplication Database [56], we used an improved method for synteny analysis. First, a BLASTP search was performed on the entire genome to identify candidate homologous gene pairs (*E*-value < 1 × 10^−5^, the first five matches). The candidate genes were analyzed with MCScanX [57] for the detection of syntenic blocks using the default parameters. MCScanX was used to distinguish singleton genes, whole-genome duplication (WGD)/segmental duplication, dispersed duplication, proximal duplication, and tandem duplication events in the strawberry *GRF* family.

### 4.6. RNA Extraction and Gene Expression Analysis

To evaluate the expression levels of *GRF* genes in strawberry, total RNA was extracted from collected plant samples or treated leaves using the RNAprep Pure Plant Plus Kit (Tiangen Biotech, Beijing, China) in accordance with the manufacturer’s recommendations. Before reverse transcription, the RNA was treated with DNase I (Tiangen Biotech) to remove residual DNA contamination. According to the RNA concentration of the sample extract, 1–7 μg of total RNA was reverse-transcribed into the first-strand cDNA using the FastKing gDNA Dispelling RT SuperMix. The reaction mixture contained 4 μL of 2 × FastKing-RT SuperMix, 4 μL of total RNA, and 12 μL of RNase-free ddH_2_O in a total reaction volume of 20 μL. The DNA polymerization temperature was 42 °C for 15 min and reverse transcriptase was deactivated at 95 °C for 3 min. Primer 5 software was used to design gene-specific primers for each *FvGRF* gene.

qRT-PCR was performed on a CFX Connect^TM^ real-time system (BIO-RAD) using SuperReal PreMix Plus (SYBR Green). Quantitative reverse transcription-PCR (qRT-PCR) reactions were performed in a 10 μL volume containing 5 μL of 2 × Super Real Pre Mix Plus, 0.3 μL of forward primers, 0.3 μL of reverse primers, 1 μL of cDNA template, and 3.4 μL of RNase-free ddH_2_O. Each amplification was conducted under the following conditions: pre-denaturation of 1 cycle for 15 min at 95 °C, followed by 40 cycles of denaturation for 10 s at 95 °C and annealing/extension for 32 s at 60 °C. The temperature was gradually increased by 0.5 °C every 10 s to analyze the melting curve. *FvPDB* [58] was used as an internal reference gene to normalize expression data. For each sample, the determination comprised three technical replicates and three biological replicates. Each replication included the internal reference gene. The relative expression level of each gene was calculated using the 2^−ΔΔ*C*t^ method, and the standard deviation was calculated from the three biological replicates and three biological replicates [59]. Sequences of primers used for qRT-PCR are listed in Appendix A.

### 4.7. Analysis of Cis-Acting Regulatory Elements in the Promoter of FvGRFs

Using the woodland strawberry reference genome, the sequence 1500 bp upstream of the start codon for each *FvGRF* gene was selected for analysis of *cis*-acting elements in the promoter using TBtools software, and the *cis*-acting elements were predicted using PlantCARE [60] (http://bionformatics.psb.ugent.be/webtools/plantcare/html (accessed on 23 March 2021)). The motifs that may be involved in plant growth and development, plant hormone responses, and abiotic and biotic stress responses were summarized.

### 4.8. Statistical Analysis

Statistical significance was determined using Student’s *t*-test as implemented in Graphpadprism7.0 software. The average ± standard deviation of at least three repeated samples was calculated. The significance of differences compared with the control were expressed as * *p* < 0.05 and ** *p* < 0.01.

## 5. Conclusions

We identified 10 *FvGRF* family members in the genome of woodland strawberry. Whole-genome duplication/segmental duplication is indicated to have been the primary driving force for expansion of the strawberry *GRF* family. The *FvGRF* genes were differentially expressed in vegetative and reproductive organs of strawberry, but were especially expressed in the roots. The genes showed varying degrees of response to low temperature, high temperature, drought, and salinity stress, and to exogenous SA and ABA treatment. The present results provide a basis for investigation of the transcriptional regulatory mechanisms of growth and development, and the functional identification of stress-resistance genes in strawberry.

## Figures and Tables

**Figure 1 plants-10-01916-f001:**
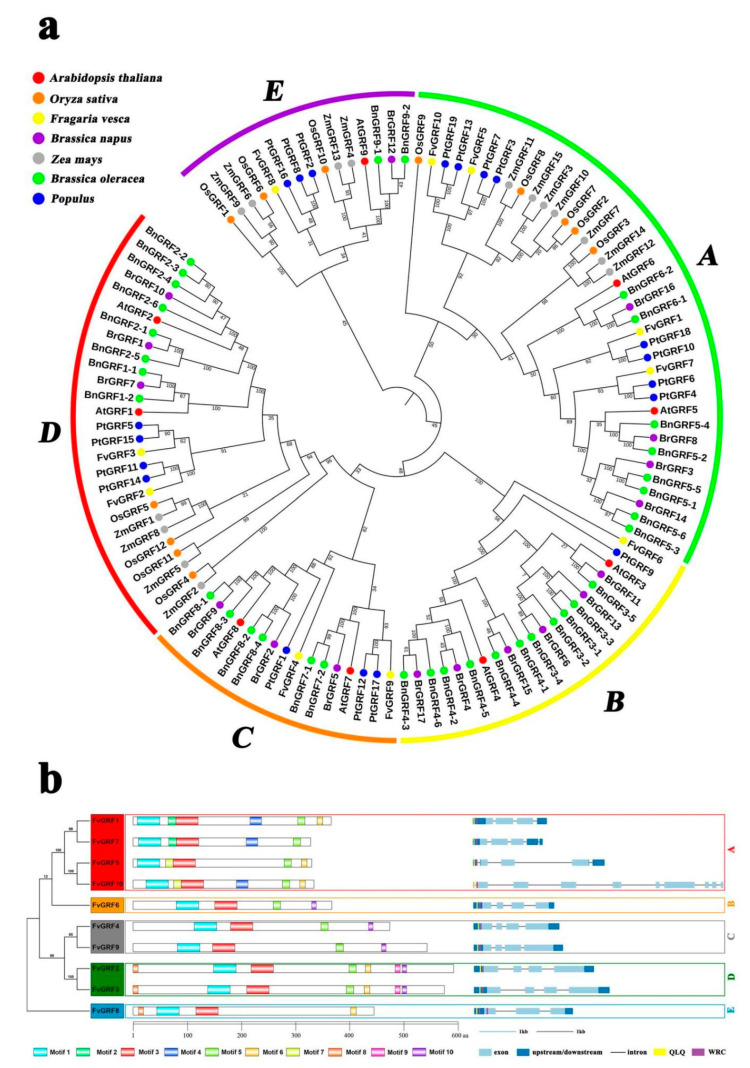
(**a**). Phylogenetic analysis of GRF proteins of Arabidopsis, rice, woodland strawberry, poplar, maize, Chinese cabbage, and rapeseed. The full-length amino acid sequences of GRF proteins from *Arabidopsis* (*AtGRF*), rice (*OsGRF*), woodland strawberry (*FvGRF*), poplar (*PtGRF*), maize (*ZmGRF*), Chinese cabbage (*BrGRF*), and rapeseed (*BnGRF*) were aligned using ClustalX. The phylogenetic tree was constructed using the maximum likelihood method with 1000 bootstrap replicates using MEGA6.0. The branches are colored to indicate *GRF* subfamilies. (**b**). Structural analysis of strawberry GRF protein. The protein domains of the strawberry *GRF* genes are shown on the left and are denoted by rectangles with different colors. The exon–intron organization is shown on the right, with exons and introns represented by light blue rectangles and black lines, respectively; UTRs are indicated by dark blue rectangles. The sequence of yellow boxes and purple boxes code the QLQ domain and WRC domain, respectively. The red, orange, gray, green, and blue rectangles are used to cluster the genes into the A, B, C, D, and E subfamilies, respectively.

**Figure 2 plants-10-01916-f002:**
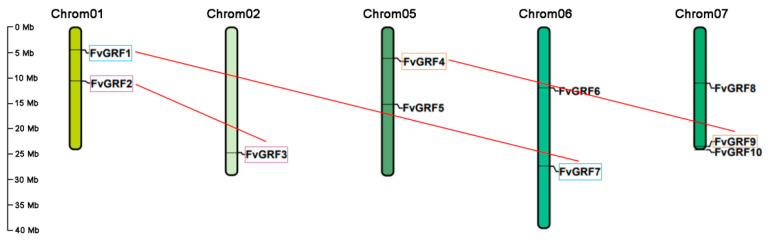
Chromosomal location of *FvGRF* genes of woodland strawberry. The chromosomal location of each *FvGRF* gene was determined in accordance with the woodland strawberry genome assembly v4.0.a1. The number of each chromosome is shown above the chromosome. Replicated fragments are represented by a colored box connected by a red line. Note: FvGRF1 (FvH4_1g08440.1), FvGRF2 (FvH4_1g18220.1), FvGRF3 (FvH4_2g32670.1), FvGRF4 (FvH4_5g10750.1), FvGRF5 (FvH4_5g23900.1), FvGRF6 (FvH4_6g18130.1), FvGRF7 (FvH4_6g34730.1), FvGRF8 (FvH4_7g12130.1), FvGRF9 (FvH4_7g32780.1), and FvGRF10 (FvH4_7g34250.1).

**Figure 3 plants-10-01916-f003:**
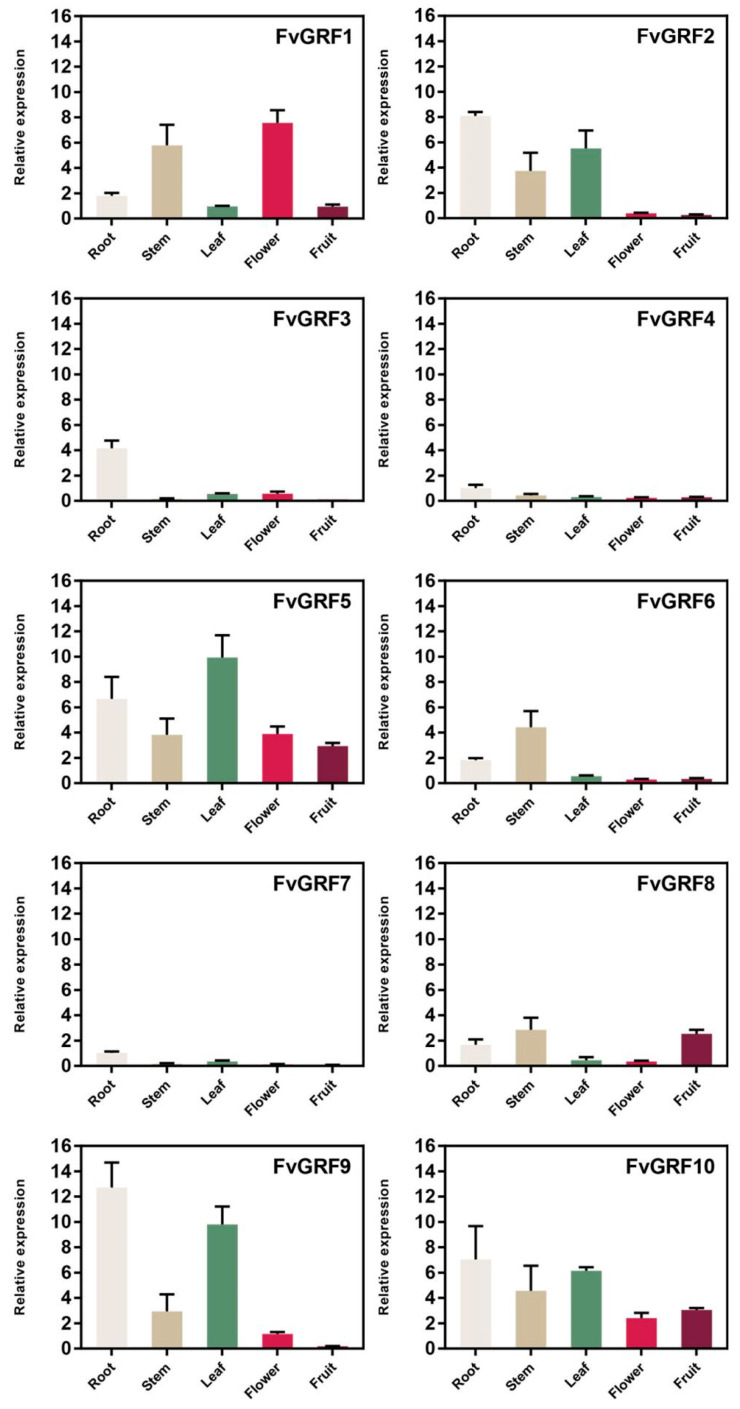
Quantitative RT-PCR analysis of 10 *FvGRF* genes in different organs of woodland strawberry. Roots, stems, and leaves are sampled during the period of vigorous growth, 42 days after the flowers are sampled, and the fruits are sampled 26 days after the flowers. Expression levels were normalized to that of *FvPDB*. The experiment was repeated three times.

**Figure 4 plants-10-01916-f004:**
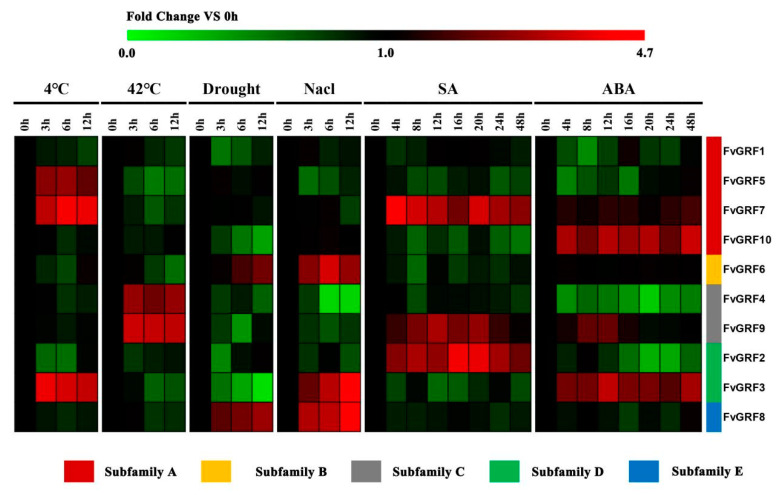
Expression accumulation profiles of 10 woodland strawberry *GRF* genes in response to low temperature (4 °C), high temperature (42 °C), drought, NaCl, salicylic acid (SA), and abscisic acid (ABA) treatments as determined by quantitative RT-PCR analysis and displayed in the form of heat maps. The color scale indicates the change in expression level, where red indicates an increase and green indicates a decrease, relative to that at 0 h. The experiment was repeated three times and the results were consistent.

**Figure 5 plants-10-01916-f005:**
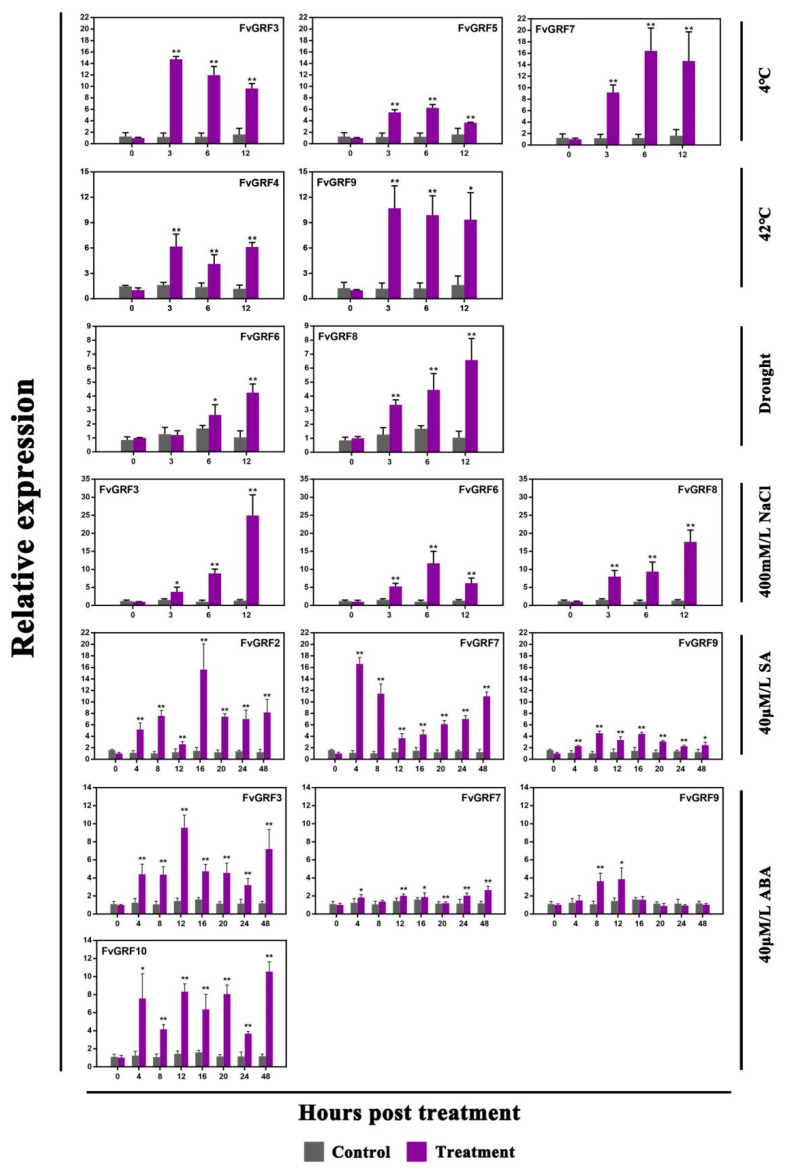
Quantitative RT-PCR analysis of woodland strawberry *GRF* gene expression in response to low temperature, high temperature, drought, NaCl, salicylic acid (SA), and abscisic acid (ABA) treatments. The expression levels of the *FvGRF* genes revealed the epigenetic patterns in response to the SA and ABA treatments. The expression levels were normalized to that of *FvPDB*. The experiment was repeated three times and consistent results were obtained. The mean and SD were calculated from three biological and three technical replicate samples. Asterisks indicate that the gene was significantly up-regulated or down-regulated after treatment (* *p* < 0.05, ** *p* < 0.01; Student’s *t*-test).

**Figure 6 plants-10-01916-f006:**
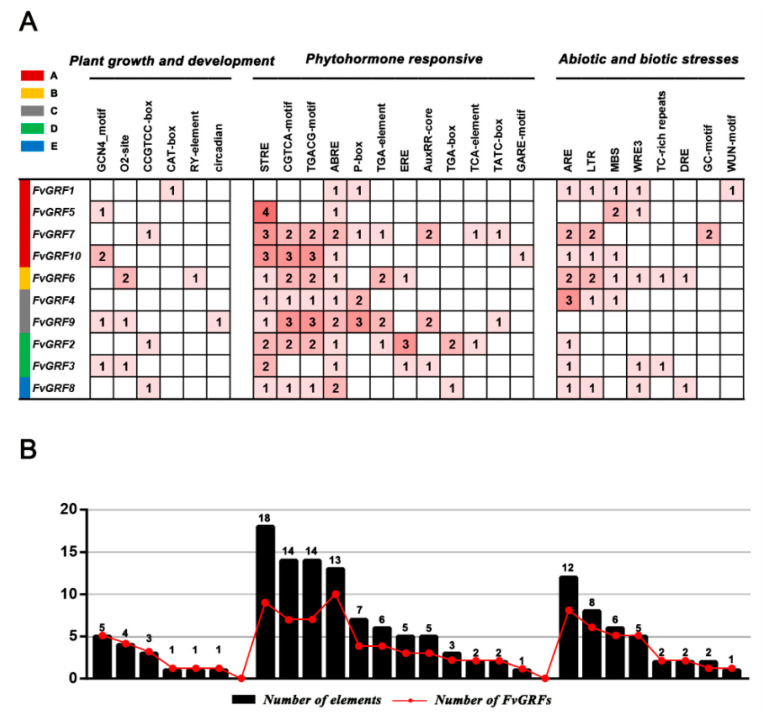
Analysis of *cis*-acting regulatory elements in the promoter of woodland strawberry *GRF* genes. (**A**) Number of each *cis*-acting element in the promoter region (1.5 kb upstream of the translation start site) of *FvGRF* genes. (**B**) Statistics for the total number of *FvGRF* genes, including the corresponding *cis*-acting element (red dot) and the total number of *cis*-acting elements in the *FvGRF* gene family (black box). On the basis of the functional annotation, the *cis*-acting elements were classified into three major classes: plant growth and development, phytohormone response, and abiotic and biotic stress response.

## Data Availability

All data in the present study are available in the public database as referred in the Materials and Methods section.

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
