# Peer review of "Genome-Wide Identification and Characterization of the Abiotic-Stress-Responsive GRF Gene Family in Diploid Woodland Strawberry (Fragaria vesca)"

_plants, 2021, doi:10.3390/plants10091916_

Round 1
Reviewer 1 Report
Li and colleagues did a routine genome-wide identification of a specific gene family in the diploid species of strawberry, in this case, the plant-specific transcription factor, growth regulatory factor or GFR. On the basis of identification of the genes, the authors analyzed the expression levels or changes of the 10 GRFs identified under various abiotic stress conditions. Regarding the procedures used in gene identification and expression analysis (qRT-PCR), the standard or the commonly used protocols were followed. Regarding data presentation, the manuscript was relatively well written. I have no issue for the manuscript to be published, but suggest the authors to deleted the content related to gene duplication analysis. Even though the results obtained based on the bioinformatics tool could be correct, interpretation of gene duplication events based on a single species does not make much sense. Instead, it should use all species from the common ancestors to strawberry and Arabidopsis to draw the conclusion, as the events detected in strawberry or Arabidopsis does not necessary mean that those events had actually happened in the individually species, rather they could have happened in different time points from the common ancestors to the divergence of strawberry and Arabidopsis.
Author Response
Response to Reviewer 1 Comments:
Hello, reviewer, first of all thank you for carefully reviewing the manuscript. Regarding the analysis of gene duplication events, we used MCsanx software to analyze Arabidopsis and strawberries, and combined with the branch classification relationship of the phylogenetic tree to judge that this result is more credible. Here, I want to solicit the opinions of reviewers and hope that this part of the content can be retained. If not, I am making corresponding changes. Thank you for reviewing again.
Reviewer 2 Report
The manuscript describes identification and characterization of 10 FvGRF family members in the genome of woodland strawberry, including gene structures and phylogenetic relationship and gene expression under different stresses. The study showed FvGRF genes were involved in plant growth, development and stress response. Although the methods are common, it could provide some useful information for plant geneticists and breeders. So I think it could be accepted after some revisions.
Suggestions:
- Please add the difference figure about the conserved domains and motifs, or gene sequnce structures. Then the Figure could be put in Figure 1 together as Figure 1b. Phylogenetic analysis as Figure 1a.
- Figure 3 could be moved to supplementary, or conbimed into Figure 1 as Figure 1c.
- Table S1 should be moved to supplementary. The gene ID could be marked in Figure 2.
- Figure 4, please add the sampling time or stages in Illustration ?
- References format is not consistency, please check.
Author Response
Response to Reviewer 2 Comments:
We greatly appreciate the thorough and thoughtful comments provided on our submitted article. We made sure that each one of the reviewer comments has been addressed carefully and the paper is revised accordingly.
We have incorporated the suggestions made by the reviewers. Those changes are highlighted within the manuscript. These changes are shown in blue in the manuscript. Below, we respond to the reviewers' comments point by point.
- We have added the conserved domain of the FvGRF gene, as shown in Figure 1b. In addition, we add the explanation of Figure 1b from line 126 to line 131. To ensure the continuity of the article, we added content from line 111 to line 120.
- We move Figure 3 to the support Additional file.
- We have put Figure S1 in the Supplementary file. In addition, we added the gene ID of the gene name in Figure 2 in lines 143 to 145.
-
In order to make the Figure 3 looks more clean and tidy,We added the tissue sampling stages in lines 173 to 175 of the article.
- We modified lines 466 to 469 to ensure the uniformity of the reference format.
- In addition, we adjusted the number of pictures and reviewed the article carefully. If you still have any questions, please inform the reviewer in time, thank you.
Reviewer 3 Report
This study presents comprehensive genetic and bioinformatic analyzes of the role of the GRF gene family in abiotic stress responses in strawberry.
The Authors used a number of bioinformatic and molecular methods, which significantly contributes to the understanding of the transcriptional regulatory mechanisms of growth and development and the functional identification of stress-resistance genes in woodland strawberry.
As a result of the research, 10 FvGRF family members have been identified, also phylogenetic analysis, chromosomal location, gene structure analysis, and conserved motif analysis have been done. Furthermore, the duplication events, expression of FvGRF genes under stress conditions and identification of cis-acting regulatory elements in the promoter of FvGRF genes have been analyzed.
Therefore, I consider the manuscript as novel and interesting and the research question as very important.
The article is clearly laid out and all the key elements are present.
The research was properly planned and performed.
The introduction provides comprehensive informations on the background to show the context of the research.
The Authors explains clearly laid out and in a logical sequence what they discovered in the research. The statistics are correct.
The claims are supported by the results. The Authors indicated how the results relate to expectations and to earlier research.
Taking the above into consideration, I recommend this manuscript for publication in present form.
Author Response
Thanks reviewer for good comments and hard work.